# Prevalence and associated factors of diabetic ketoacidosis among patients with diabetes mellitus at the University of Gondar Comprehensive and Specialized Referral Hospital Northwest, Ethiopia

Abebe Birhanu[1]*, Sintayehu Ambachew[2], Netsanet Baye[3], Emiyamrew Getnet[3], Sintayehu Admas[4], Eshet Gebrie[2], Abebaw Worede[2]

1 Department of Medical Microbiology, School of Biomedical and Laboratory Sciences, College of Medicine and Health Sciences, University of Gondar, Gondar, Ethiopia, 2 Department of Clinical Chemistry, School of Biomedical and Laboratory Sciences, College of Medicine and Health Sciences, University of Gondar, Gondar, Ethiopia, 3 Department of Immunology and Molecular Biology, School of Biomedical and Laboratory Sciences, College of Medicine and Health Sciences, Wollo University, Wollo, Ethiopia, 4 Department of Hematology and Immunohematology, School of Biomedical and Laboratory Sciences, College of Medicine and Health Sciences, University of Gondar, Gondar, Ethiopia

* Abebe.Birhanu@uog.edu.et, abebebir10@gmail.com

## Abstract

### Background

Diabetes is a group of metabolic diseases characterized by hyperglycemia resulting from defects in insulin secretion, insulin action, or both. Diabetic ketoacidosis is one of the life-threatening complications in diabetic individuals with, high morbidity and mortality globally. However, the data related to the prevalence and associated factors of diabetic ketoacidosis are limited in the study setting.

### Objective

To assess the prevalence of diabetic ketoacidosis and its associated factors among diabetic mellitus patients at the University of Gondar Comprehensive Specialized Hospital.

### Methods

A hospital-based cross-sectional study was conducted from March 1 to September 30, 2021. A total of 405 diabetic patients aged 20 and above were selected using a systematic random sampling technique. A total of 810 blood and urine samples (each 405) were collected using sterile serum separator tubes and urine collection cups, respectively. Sociodemographic and clinical data was collected using a structured questionnaire. Chemical analysis of urine was done using urine reagent strips to determine urine ketone bodies and $P^H$. BECKMAN COULTER DxC700 AU clinical chemistry analyzer instrument was used to determine electrolytes and metabolites. The data was entered using Epi-Data version 4.6 and transferred to SPSS version 25 for analysis. Bivariable and multivariable logistic

**Data Availability Statement:** All relevant data are within the paper.

**Funding:** The author(s) received no specific funding for this work.

**Competing interests:** The authors have declared that no competing interests exist.

**Abbreviations:** AOR, Adjusted Odds Ratio; CI, Confidence Interval; COR, Crude Odds Ratio; DKA, Diabetic Ketoacidosis; DM, Diabetes Mellitus; NPH, Neutral Protamine Hagedorn; OHAs, Oral Hypoglycemic Agents; T1DM, Type I Diabetes Mellitus; T2DM, Type II Diabetes Mellitus; UoGCSH, University of Gondar Comprehensive Specialized Hospital.

regression analyses were used to determine the factors associated with the diabetic ketoacidosis. The results were considered statistically significant if the adjusted odds ratio was reported with a 95% confidence interval and a $P$-value below 0.05.

## Results

The overall prevalence of diabetic ketoacidosis among diabetic patients was 35/405 (8.6%, 95% CI: 6.0–11.0%). Of these cases, 25 (71.4%) had type 1 diabetes mellitus, while 10 (28.6%) had type 2 diabetes mellitus. Statistically significant factors associated with diabetic ketoacidosis included being a young adult aged 20–29 years (AOR = 2.262; 95% CI = 1.090–4.758; $P$ = 0.013), unemployment (AOR = 2.578; 95% CI = 1.457–6.113; $P$ = 0.017), the presence of infection (AOR = 2.819; 95% CI = 1.138–8.428; $P$ = 0.024), and being T1DM (AOR = 3.106; 95% CI = 1.150–7.273; $P$ = 0.003).

## Conclusions and recommendations

The prevalence of diabetic ketoacidosis among follow-up diabetes patients in this study was high, particularly among those aged 20–29 years, unemployed, or with infections. Increased vigilance, regular monitoring, timely infection management, and comprehensive diabetes education are essential for early detection and prevention of DKA. Social and financial support for unemployed diabetic patients can further enhance access to care and reduce DKA risk.

## Introduction

Diabetes mellitus (DM) is a metabolic disorder characterized by the body's inability to effectively utilize glucose, leading to its accumulation in the bloodstream. This results from abnormalities in processes such as gluconeogenesis and glycogenolysis [1]. Diabetes mellitus is classified into type 1 diabetes mellitus (T1DM), type 2 diabetes mellitus (T2DM), gestational diabetes mellitus (occurring during pregnancy), and other specific types [2]. Type I diabetes mellitus is an autoimmune condition where the destruction of pancreatic beta cells causes a complete lack of insulin production. In contrast, T2DM develops due to a combination of reduced insulin secretion and insulin resistance. This resistance primarily affects the liver and muscles, increases glucose production in the liver, releases excessive free fatty acids from adipose tissue, and leads to a gradual decline in beta cell function [3, 4].

In 2019, DM affected approximately 463 million adults aged 20 to 79 worldwide, with this number projected to increase to 578 million by 2030 and 700 million by 2045 [5]. Globally, DM and its complications caused 4.2 million deaths among adults aged 20 to 99 and resulted in an estimated $760 billion in medical expenses [6]. In sub-Saharan Africa, DM impacted over 12 million people in 2020, contributing to approximately 330,000 deaths [7]. The prevalence of DM in the region has been steadily increasing, with forecasts suggesting that by 2045, the number of cases will rise to about 40.7 million, compared to 15.9 million reported in 2017 [8].

Diabetic ketoacidosis (DKA) is a serious complication of DM that primarily affects individuals with T1DM due to insufficient insulin, leading to high blood sugar levels and excessive ketone production. However, it can also develop in individuals with T2DM [9, 10]. This critical condition stems from an insulin deficiency caused by the autoimmune destruction of

pancreatic beta cells, resulting in decreased insulin activity, increased insulin demand, and elevated levels of counter-regulatory hormones [11]. Consequently, glucose cannot enter cells, forcing them to rely on free fatty acids for energy due to intracellular starvation [10]. These fatty acids are transported to the liver, where they undergo oxidation in the mitochondria, producing ketone bodies such as acetone, acetoacetate, and beta-hydroxybutyrate [12]. The accumulation of these acidic ketones leads to ketonemia and metabolic acidosis, which are defining characteristics of DKA. Additionally, high blood glucose levels induce osmotic diuresis, causing significant fluid and electrolyte loss. Without prompt rehydration, this dehydration can impair kidney function, reducing the glomerular filtration rate and worsening the metabolic imbalances associated with DKA [11].

Diabetic ketoacidosis has a mortality rate ranging from 2–5% in high-income countries to 6–24% in low-income nations when timely diagnosis and adequate treatment are not provided [10]. In Ethiopia, DKA-related deaths have been reported at 11–12% [13]. Various factors can trigger DKA, including infections such as urinary tract infections and pneumonia, poor adherence to diabetes medications, myocardial infarction, stroke, acute pancreatitis, trauma, burns, and surgical procedures [14]. However, many DKA cases can be effectively managed with improved access to healthcare, including intravenous rehydration, continuous insulin therapy, and correction of electrolyte imbalances [15].

The prevalence of DKA varies widely across different regions, including within Ethiopia, with most studies reporting a high occurrence rate [14]. However, there is a lack of primary data on the prevalence and contributing factors of DKA among diabetic patients receiving follow-up care at the University of Gondar Comprehensive Specialized Hospital (UoGCSH) in Northwest Ethiopia. This study, therefore, was conducted to determine the prevalence of DKA and identify its associated factors among diabetic patients at UoGCSH. The results will provide essential information for health program administrators to design effective strategies for managing and preventing DKA in diabetic patients. Additionally, the study aims to increase community awareness of the risk factors associated with DKA and establish a recent baseline for future research in this field.

## Methods and materials

### Study design, period, and area

A hospital-based cross-sectional study was conducted on follow-up diabetic patients from March 1 to September 30, 2021, at the UoGCSH in Northwest Ethiopia. The UoGCSH, a hospital with 977 beds spread across 29 wards and emergency rooms, serves a population of approximately 13 million people from the surrounding and neighboring regions. It provides a wide range of medical services, including internal medicine, surgery, obstetrics and gynecology, pediatrics, laboratory diagnostics, eye care, physiotherapy, dental care, cervical health, psychiatry, dermatology, and medication distribution. Additionally, the hospital offers social services and houses specialized units for treating tuberculosis, kala-azar, cancer, fistula surgery, psychiatric and psychological care, palliative and rehabilitation services, intensive care for adults and children, and cataract surgery [16].

### Study population

The source population for the study was all DM patients who had regular follow-up at the diabetes clinic of UoGCSH, while the study population was DM patients who visited the diabetes clinic of UoGCSH during the study period. The study included all adult DM patients aged 18 and above who had regular follow-up at the diabetes clinic of UoGCSH. However, pregnant and breastfeeding women were excluded from the study.

## Sample size determination and sampling technique

The sample size for this study was calculated using the single population proportion formula. The calculation assumed a 40% prevalence of DKA among DM patients in Ethiopia, Hawassa, a 95% confidence interval (CI), and a 5% margin of error. The formula is as follows:

$$n = \frac{(Z_{a/2})^2 * p(1-p)}{d^2}$$

Where:

n is the minimum required sample size.

$Z\alpha/_2$ is the Z-score for a 95% CI, which equals 1.96.

$p$ is the prevalence of DKA ($p$ = p = 0.4).

d is the margin of error ($d$ = 0.05).

By substituting these values:

n = $(1.96)^2$ x (0. 4 x0.6)/ $(0.05)^2$

n = 0.921984/0.0025 = 369

To account for a 10% non-response rate, the sample size was adjusted to 405 participants (n = 369+36 = 405). A systematic random sampling technique was used to select the 405 study participants. According to 2020 hospital records, about 60 DM patients visited UoGCSH daily. With 20 working days per month over seven months, the total number of DM patients was estimated to be 8400. The sampling interval was calculated as 8400/405 = 21. The first participant on the initial day of sample collection was chosen using a lottery method. Subsequently, every 21st patient was invited to participate until the required sample size was achieved. This process was applied monthly, with the following calculations for the $K^{th}$ value for each month as follows (Fig 1).

$K^{th}$ value for DM patients in March = 1400 / 67 = 20.8≈21

$K^{th}$ value for DM patients in April = 1200 / 58 = 20.7≈21

$K^{th}$ value for DM patients in May = 1300 / 63 = 20.6≈21

$K^{th}$ value for DM patients in June = 1200 / 58 = 20.7≈21

$K^{th}$ value for DM patients in July = 1100 / 53 = 20.8≈21

$K^{th}$ value for DM patients in August = 1000 /48 = 20.8≈21

$K^{th}$ value for DM patients in September n = 1200 / 58 = 20.7≈21

Proportional allocation was used to ensure the sample size matched patient flow during each month. The number of patients sampled per month was calculated using the formula:

Nf = Average no of patients in each Month × total sample size

Total follow up diabetes patients in diabetes clinic

The resulting monthly allocations were (Fig 1):

March (n = (1400 x 405)/ 8400 = 67)

April (n = (1200 x 405)/ 8400 = 58)

May (n = (1300 x 405)/ 8400 = 63)

June (n = (1200 x 405)/ 8400 = 58)

July (n = (1100 x 405)/ 8400 = 53)

August (n = (1000 x 405)/ 8400 = 48)

September (n = (1200 x 405)/ 8400 = 58)

## Variables of the study

The study encompassed various variables, categorized into outcome and predictive variables. The outcome variable was DKA. Predictive variables included sociodemographic characteristics of the participants, such as age, sex, educational level, occupational status, place of

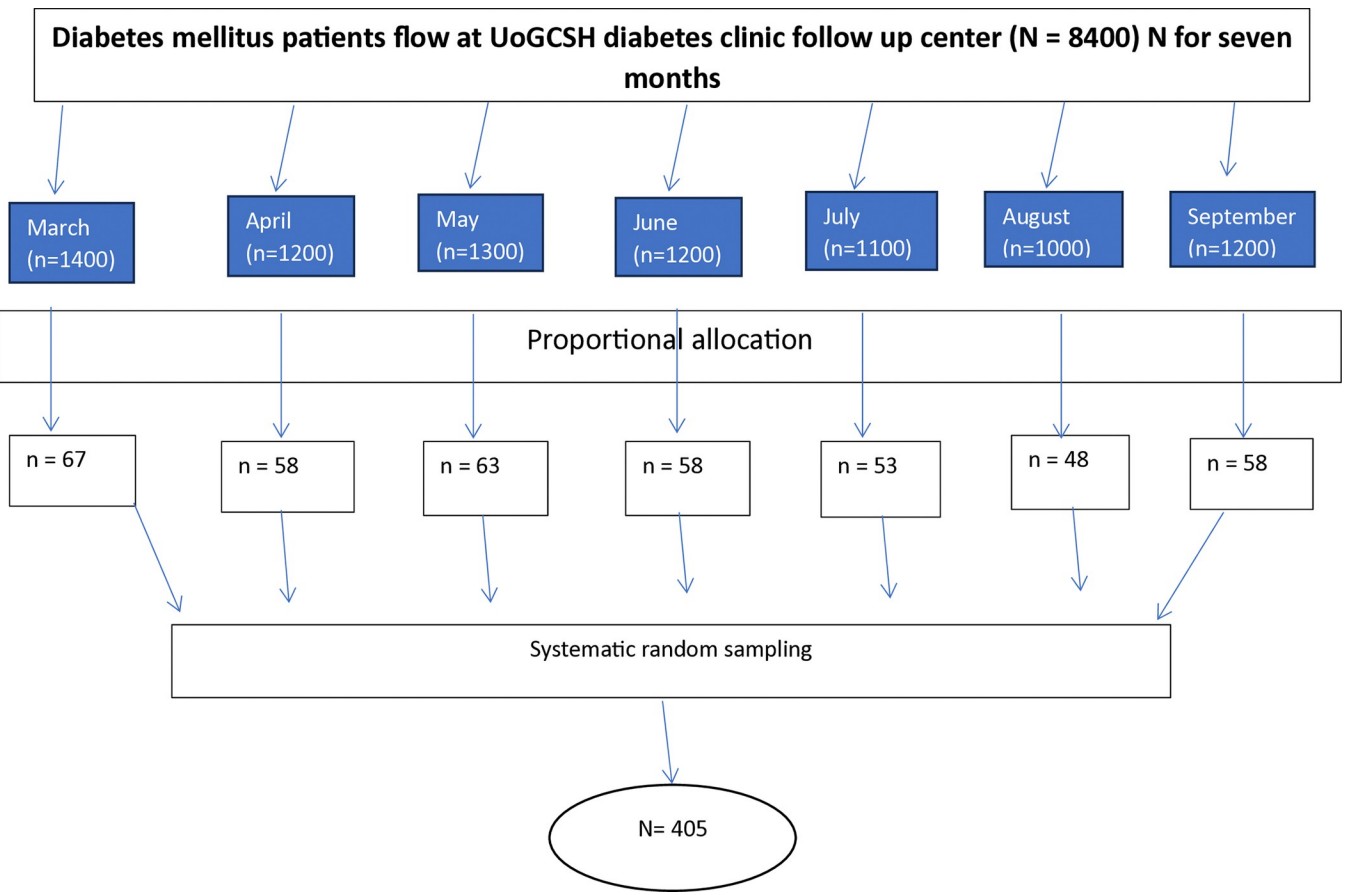

**Fig 1. The diagrammatic presentation of sampling procedure and patient flow of diabetes patients in diabetes clinic of UoGCSH, Northwest Ethiopia, from March 1 to September 30, 2021.**

residence, monthly income, and marital status. Lifestyle and behavioral factors of DM patients were also considered, including physical exercise, smoking, khat chewing, alcohol consumption, and coffee drinking. Additionally, clinical information related to DM was examined, covering hypertension status, type and duration of diabetes, treatment methods, family history of diabetes, presence of infections, body mass index (BMI), and the coexistence of other chronic conditions.

## Data collection

Skilled nurses gathered sociodemographic and clinical information from DM patients through face-to-face interviews, utilizing a structured questionnaire to ensure consistency and accuracy.

## Sample collection and processing

Trained laboratory technologists meticulously adhered to standardized protocols for sample collection. After disinfecting the puncture site with 70% alcohol, they used sterile 21-gauge disposable syringes to draw 5 mL of blood from the median cubital vein of each of the 405 DM patients, depositing the samples into serum separator tubes. Additionally, 10 mL of urine was

collected from each participant using sterile urine collection cups for further laboratory investigation.

After collection, the blood samples were centrifuged at 3,500 revolutions per minute for 10 minutes to separate the serum. The serum was then analyzed using the BECKMAN COULTER DxC700 AU clinical chemistry analyzer to measure fasting blood glucose levels and electrolytes, including sodium ($[Na^+]$), chloride ($[Cl^-]$), and bicarbonate ($[HCO_3^-]$) concentrations. Urine reagent strips were used to test the urine samples for chemical parameters such as pH, protein, glucose, and ketone bodies.

### Operational definitions

Diabetic ketoacidosis is characterized by fasting blood glucose levels of $\geq$200–250 mg/dL, the presence of ketones in the urine ($\geq$+2), an anion gap >10–12 mEq/L, serum bicarbonate <15–18 mEq/L, and a pH range of <7.0–7.3 [17, 18].

Fasting blood glucose refers to the measurement of blood sugar levels obtained from venous blood after at least 8 hours of overnight fasting [19].

Obesity is defined as a BMI of $\geq$30 kg/m$^2$, while overweight is classified as a BMI between 25 and 29.9 kg/m$^2$. Individuals with a BMI <18.5 kg/m$^2$ are categorized as underweight, and those with a BMI of 18.5–24.9 kg/m$^2$ are considered to have a normal weight [20, 21].

Diabetic patients are defined as individuals who have been previously diagnosed with DM.

### Data analysis

The data was reviewed for completeness, coded, and then entered into Epi-Data version 4.6. Subsequently, it was exported and analyzed using the Statistical Package for Social Sciences (SPSS) version 25. Descriptive analysis was employed to calculate the frequency and percentage of variables. Independent variables with a $P$-value $\leq$ 0.2 in the bivariate analysis were included in a multivariate logistic regression model to simultaneously account for potential confounding factors. Statistical significance was determined by an adjusted odds ratio with a $P$-value < 0.05 at a 95% CI. The findings were presented using narrative descriptions, graphs, and tables.

### Ethical approval

The Institutional Review Committee of the School of Biomedical and Laboratory Sciences, College of Medicine and Health Sciences, University of Gondar, approved the study under reference protocol number SBMLS/2453. Prior to data collection, written informed consent was obtained from all study participants. Participants were thoroughly informed about the purpose and procedures of the research to ensure they fully understood before providing their consent. Additionally, all data and samples collected were handled with strict confidentiality, securely stored, and used solely for the purposes specified in the study protocol, thereby safeguarding the privacy and anonymity of the participants.

## Results

### Sociodemographic characteristics of diabetic mellitus patients

A total of 405 follow-up DM patients were included in this study, with a 100% response rate. Among these, 207 study subjects were males (51.1%). The mean (± SD) age of study participants was 50.8 (± 15.6) years. The majority of them belonged to the age group 50–59 years (25.2%), were married (52.1%), and lived in urban area (79%) (Tables 1 and 2).

**Table 1. Sociodemographic characteristics of diabetes mellitus patients at the UoGCSH from February 15 to September 30, 2021.**

| Characteristics of study participants | Category | Frequency N (%) | Diabetic ketoacidosis | | Fisher's Exact Test |
|---|---|---|---|---|---|
| | | | Yes N (%) | No N (%) | P-value |
| Sex | Male | 207 (51.1) | 25 (12.1) | 182 (87.9) | 0.460 |
| | Female | 198 (48.9) | 10 (5.1) | 188 (94.9) | |
| Age (years) | 20–29 | 52 (12.8) | 17 (32.7) | 35 (67.3) | **0.013**[*] |
| | 30–39 | 59 (14.6) | 8 (13.6) | 51 (86.4) | |
| | 40–49 | 68 (16.8) | 5 (7.4) | 63 (92.6) | |
| | 50–59 | 102 (25.2) | 3 (2.9) | 99 (97.1) | |
| | 60–69 | 79 (19.5) | 1 (1.3) | 78 (98.7) | |
| | ≥70 | 45 (11.1) | 1 (2.2) | 44 (97.8) | |
| Residence | Urban | 320 (79.0) | 20 (6.3) | 300 (93.7) | 0.145 |
| | Rural | 85 (21.0) | 15 (17.6) | 70 (82.4) | |
| Educational status | Illiterate | 147 (36.3) | 13 (8.8) | 134 (91.2) | 0.677 |
| | Elementary completed | 94 (23.2) | 11 (11.7) | 83 (88.3) | |
| | Secondary completed | 57 (14.1) | 8 (14.0) | 49 (86.0) | |
| | Higher education | 107 (26.4) | 3 (2.8) | 104 (97.2) | |
| Occupational status | Governmental | 52 (12.8) | 2 (3.8) | 50 (96.2) | **0.039**[*] |
| | Merchant | 71 (17.5) | 8 (11.3) | 63 (88.7) | |
| | Housewife | 101 (24.9) | 8 (7.9) | 93 (92.1) | |
| | Farmer | 98 (24.2) | 7 (7.1) | 91 (92.9) | |
| | Unemployed | 83 (20.5) | 10 (12.0) | 73 (88.0) | |
| Income of diabetes patients (ETB) | < 500 | 142 (35.1) | 22 (15.5) | 120 (84.5) | 0.246 |
| | 501–1000 | 86 (21.2) | 7 (8.1) | 79 (91.9) | |
| | 1001–1500 | 39 (9.6) | 2 (5.1) | 37 (94.9) | |
| | >1500 | 138 (34.1) | 4 (2.9) | 134 (97.1) | |
| Marital status | Married | 211 (52.1) | 15 (7.1) | 196 (92.9) | 0.102 |
| | Single | 82 (20.2) | 7 (8.5) | 75 (91.5) | |
| | Divorced | 49 (12.1) | 7 (14.3) | 42 (85.7) | |
| | Windowed | 17 (4.2) | 3 (17.6) | 14 (82.4) | |
| | Separated | 46 (11.4) | 3 (6.5) | 43 (93.5) | |

[*] The observed difference is statistically significant at P-value < 0.05, ETB = Ethiopian birr

### Clinical and lifestyle (behavioural) characteristics of diabetic mellitus patients

The mean duration of diabetes among DM patients was 6.8 years, with a standard deviation of 5.4 years. Most of the study participants had T2DM (62.5%), a habit of coffee drinking (70.9%), a normal BMI (72.0%), and no family history of diabetes (77.0%) (Tables 3 and 4).

### Laboratory findings of diabetic mellitus patients

Most of the study participants had urine pH <7 (94.8%), serum bicarbonate ≥15 mEq/L (84.7%), and serum anion gap ≤12 mEq/L (Tables 5 and 6).

### Prevalence of diabetic ketoacidosis

Of the total, 35/405 (8.6%) (95% CI: 6.0–11.0%) of the participants were presented with DKA. Among these, T1DM with DKA was 25/35 (71.4%) and T2DM with DKA was 10/35 (28.6%) (Fig 2).

**Table 2. Distribution of sociodemographic characteristics by diabetes type among patients at the UoGCSH, Northwest Ethiopia, from March 1 to September 30, 2021.**

| Characteristics of study participants | Category | Frequency N (%) | DM type | | Fisher's Exact Test |
|---|---|---|---|---|---|
| | | | T1DM N (%) | T2DM N (%) | P-value |
| Sex | Male | 207 (51.1) | 78 (37.7) | 129 (62.3) | 0.888 |
| | Female | 198 (48.9) | 74 (37.4) | 124 (62.6) | |
| Age (years) | 20–29 | 52 (12.8) | 42 (80.8) | 10 (19.2) | **<0.001**[*] |
| | 30–39 | 59 (14.6) | 40 (67.8) | 19 (32.2) | |
| | 40–49 | 68 (16.8) | 25 (36.8) | 43 (63.2) | |
| | 50–59 | 102 (25.2) | 24 (23.5) | 78 (76.5) | |
| | 60–69 | 79 (19.5) | 14 (17.7) | 65 (82.3) | |
| | ≥70 | 45 (11.1) | 7 (15.6) | 38 (84.4) | |
| Residence | Urban | 320 (79.0) | 100 (31.3) | 220 (68.7) | **0.029**[*] |
| | Rural | 85 (21.0) | 52 (61.2) | 33 (38.8) | |
| Educational status | Illiterate | 147 (36.3) | 50 (34.0) | 97 (66.0) | 0.522 |
| | Elementary completed | 91 (23.2) | 33 (36.3) | 58 (63.7) | |
| | Secondary completed | 57 (14.1) | 27 (47.4) | 30 (52.6) | |
| | Higher education | 110 (26.4) | 42 (38.2) | 68 (61.8) | |
| Occupational status | Governmental | 52 (12.8) | 17 (32.7) | 35 (67.3) | **0.012**[*] |
| | Merchant | 71 (17.5) | 23 (32.4) | 48 (67.6) | |
| | Housewife | 101 (24.9) | 37 (36.6) | 64 (63.4) | |
| | Farmer | 98 (24.2) | 48 (49.0) | 50 (51.0) | |
| | Unemployed | 83 (20.5) | 27 (32.5) | 56 (67.5) | |
| Income of diabetes patients (ETB) | < 500 | 142 (35.1) | 67 (47.2) | 75 (52.8) | 0.627 |
| | 501–1000 | 86 (21.2) | 29 (33.7) | 57 (66.3) | |
| | 1001–1500 | 39 (9.6) | 6 (15.4) | 33 (84.6) | |
| | >1500 | 138 (34.1) | 50 (36.2) | 88 (63.8) | |
| Marital status | Married | 291 (52.1) | 96 (33.0) | 195 (67.0) | **0.010**[*] |
| | Single | 42 (20.2) | 31 (73.8) | 11 (26.2) | |
| | Divorced | 49 (12.1) | 16 (32.7) | 33 (67.3) | |
| | Windowed | 17 (4.2) | 7 (41.2) | 10 (58.8) | |
| | Separated | 6 (11.4) | 2 (33.3) | 4 (66.7) | |

[*] The observed difference is statistically significant at P-value < 0.05, ETB = Ethiopian birr

A high frequency of DKA was also observed among participants within the 20–29 age group, 17/35 (48.6%) (Fig 3).

### Associated factors of diabetic ketoacidosis

Diabetes mellitus patients within the age group of 20–29 years (AOR = 2.262; 95%CI = 1.090–4.758; P = 0.013), being unemployed in occupation (AOR = 2.578; 95%CI = 1.457–6.113; P = 0.017), presence of infection (AOR = 2.819; 95%CI = 1.138–8.428; P = 0.024), and being T1DM type (AOR = 3.106; 95%CI = 1.150–7.273; P = 0.003) had showed statistically significant association with DKA. However, none of the remaining variables showed significant association with DKA in this study. The odds of DKA among the study participants were higher in younger than older study participants; study participants in the age group of 20–29 were 2.262 times at higher risk of developing DKA than those with seventy and above. Unemployment in occupational status was another factor significantly associated with the presence of DKA in such, unemployed DM patients were 2.578 times more likely to develop DKA as compared to

**Table 3. Clinical and lifestyle (behavioural) characteristics of diabetic mellitus patients at the UoGCSH, Northwest Ethiopia, from March 1 to September 30, 2021.**

| Characteristics of study participants | Category | Frequency N (%) | Diabetic ketoacidosis | | Fisher's Exact Test |
|---|---|---|---|---|---|
| | | | Yes N (%) | No N (%) | P-value |
| Physical exercise | Yes | 147 (36.3) | 15 (10.2) | 132 (89.8) | 0.463 |
| | No | 258 (63.7) | 20 (7.8) | 238 (92.2) | |
| Hypertension | Yes | 154 (38.0) | 9 (5.8) | 145 (94.2) | 0.691 |
| | No | 251 (62.0) | 26 (10.4) | 225 (89.6) | |
| Smoking cigarette | Yes | 6 (1.5) | 2 (33.3) | 4 (66.7) | 0.087 |
| | No | 399 (98.5) | 33 (8.3) | 366 (91.7) | |
| Chat chewing | Yes | 5 (1.2) | 0 (0) | 5 (100.0) | 0.635 |
| | No | 400 (98.8) | 35 (8.8) | 365 (91.2) | |
| Alcohol use | Yes | 82 (20.2) | 12 (14.6) | 70 (85.4) | 0.564 |
| | No | 323 (79.8) | 23 (7.1) | 300 (92.9) | |
| Coffee drinking | Yes | 287 (70.9) | 24 (8.4) | 263 (91.6) | 0.846 |
| | No | 118 (29.1) | 11 (9.3) | 107 (90.7) | |
| Type of diabetes mellitus | Type I | 152 (37.5) | 25 (16.4) | 127 (83.6) | **0.002**[*] |
| | Type II | 253 (62.5) | 10 (4.0) | 243 (96.0) | |
| Duration of diabetes | <5 | 179 (44.2) | 14 (7.8) | 165 (92.2) | 0.828 |
| | 5–15 | 193 (47.7) | 18 (9.3) | 175 (90.7) | |
| | >15 | 33 (8.1) | 3 (9.1) | 30 (90.9) | |
| Types of diabetes treatment | NPH insulin | 152 (37.5) | 25 (16.4) | 127 (83.6) | **0.045**[*] |
| | OHA | 230 (56.8) | 8 (3.5) | 222 (96.5) | |
| | Both NPH and OHA | 23 (5.7) | 2 (8.7) | 21 (91.3) | |
| Family history of diabetes | Yes | 93 (23.0) | 9 (9.7) | 84 (90.3) | 0.278 |
| | No | 312 (77.0) | 26 (8.3) | 286 (91.7) | |
| Presence of infection | Yes | 155 (38.3) | 6 (3.9) | 149 (96.1) | **0.006**[*] |
| | No | 250 (61.7) | 29 (11.6) | 221 (88.4) | |
| Presence of additional chronic diseases | Yes | 94 (23.2) | 5 (5.3) | 89 (94.7) | 0.216 |
| | No | 311 (76.8) | 30 (9.6) | 281 (90.4) | |
| BMI (kg/m2) | Normal | 292 (72.0) | 28 (9.6) | 264 (90.4) | 0.439 |
| | Underweight | 8 (2.0) | 1 (12.5) | 7 (87.5) | |
| | Overweight | 93 (23.0) | 5 (5.4) | 88 (94.6) | |
| | Obese | 12 (3.0) | 1 (8.3) | 11 (91.7) | |

[*]The observed difference is statistically significant at P-value < 0.05, BMI = body mass index, OHA = oral hypoglycemic agents, NPH = neutral protamine Hagedorn

participants who have governmental occupational status. In addition, DM patients with infection had a 2.819 times greater chance of developing DKA than those without infections. Furthermore, T1DM was 3.106 times more likely to develop DKA than T2DM (Table 7).

## Discussion

In the current study, the prevalence of DKA was 8.6% (95% CI: 6.0–11.0%), which is consistent with findings from a study in Malaysia (8.2%) [22]. However, the prevalence of this study is higher than reported rates from studies conducted in Nigeria (0.13%) [23], Ghana (2.0%) [24], Tanzania (1.5%) [25], and Burkina Faso (5.2%) [26]. The observed difference may be associated with differences in the geographical area, sample size, diabetes type, health-seeking behaviour, dietary habits, and general lifestyle of patients [24, 27]. For example, a study conducted in Ghana [24] specifically examined individuals with T2DM and did not include those with T1DM. In contrast, our research considered T1DM besides to T2DM, with T1DM patients

**Table 4. Distribution of behavioural characteristics by diabetes type among patients at the UoGCSH, Northwest Ethiopia, from March 1 to September 30, 2021.**

| Characteristics of study participants | Category | Frequency N (%) | DM type | | Fisher's Exact Test |
|---|---|---|---|---|---|
| | | | T1DM N (%) | T2DM N (%) | P-value |
| Physical exercise | Yes | 147 (36.3) | 65 (44.2) | 82 (55.8) | **0.043**[*] |
| | No | 258 (63.7) | 87 (33.7) | 171 (66.3) | |
| Hypertension | Yes | 154 (38.0) | 40 (26.0) | 114 (74.0) | **<0.001**[*] |
| | No | 251 (62.0) | 112 (44.6) | 139 (55.4) | |
| Smoking cigarette | Yes | 6 (1.5) | 2 (33.3) | 4 (66.7) | **0.047**[*] |
| | No | 399 (98.5) | 150 (37.6) | 249 (62.4) | |
| Chat chewing | Yes | 5 (1.2) | 3 (60.0) | 2 (40.0) | 0.368 |
| | No | 400 (98.8) | 149 (37.2) | 251 (62.8) | |
| Alcohol use | Yes | 82 (20.2) | 34 (41.5) | 48 (58.5) | 0.444 |
| | No | 323 (79.8) | 118 (36.5) | 205 (63.5) | |
| Coffee drinking | Yes | 287 (70.9) | 106 (36.9) | 181 (63.1) | 0.735 |
| | No | 118 (29.1) | 46 (39.0) | 72 (61.0) | |
| Duration of diabetes | <5 | 179 (44.2) | 61 (34.1) | 118 (65.9) | 0.085 |
| | 5–15 | 193 (47.7) | 73 (37.8) | 120 (62.2) | |
| | >15 | 33 (8.1) | 18 (54.5) | 15 (45.5) | |
| Types of diabetes treatment | NPH insulin | 151 (37.5) | 151 (100.0) | 0 (0) | **0.001**[*] |
| | OHA | 230 (56.8) | 1 (0.4) | 229 (99.6) | |
| | Both NPH and OHA | 24 (5.7) | 0 (0) | 24 (100.0) | |
| Family history of diabetes | Yes | 93 (23.0) | 37 (39.8) | 56 (60.2) | **0.040**[*] |
| | No | 312 (77.0) | 115 (36.9) | 197 (63.1) | |
| Presence of infection | Yes | 155 (38.3) | 46 (29.7) | 109 (70.3) | **0.002**[*] |
| | No | 250 (61.7) | 106 (42.4) | 144 (57.6) | |
| Presence of additional chronic diseases | Yes | 94 (23.2) | 29 (30.9) | 65 (69.1) | 0.145 |
| | No | 311 (76.8) | 123 (39.5) | 188 (60.5) | |
| BMI (kg/m2) | Normal | 292 (72.0) | 116 (39.7) | 176 (60.3) | **0.022**[*] |
| | Underweight | 8 (2.0) | 5 (62.5) | 3 (37.5) | |
| | Overweight | 93 (23.0) | 27 (29.0) | 66 (71.0) | |
| | Obese | 12 (3.0) | 4 (33.3) | 8 (66.7) | |

[*]The observed difference is statistically significant at P-value < 0.05, BMI = body mass index, OHA = oral hypoglycemic agents, NPH = neutral protamine Hagedorn

typically being more prone to DKA, potentially explaining the higher prevalence in our findings. On the other hand, the DKA prevalence observed in our study is lower than rates reported in several studies from Ethiopia, including those in Woldiya (31%) [28], Dessie (19.9%) [29], Mekelle (78.7%) [30], Adama (66.5%) [31], Dila (38.0%) [32], Gurage (14.2%) [33], Harar (67.6%) [34], Hawassa (40.0%) [27], Jimma (19.5%, 33.7%, 49.2%, 92.6%) [35–38], and Addis Ababa (20.8%, 35.8%) [39, 40], as well as studies from Gambia (22.3%) [41], Egypt (18.5%, 53.5%) [42, 43], Sudan (17.6%) [44], Benin (21.4%) [45], Nigeria (12.2%) [46], Sweden (22.8%) [47], Serbia (35.1%) [48], Italy (36.9%, 41.9%) [49, 50], Iraq (40.0%) [51], and Germany (19.8%) [52]. The differences in DKA prevalence between studies could stem from factors such as differences in study design, the age distribution of participants, the specific type of diabetes examined, methods of glucose control, and variations in healthcare access and patient education [29]. For example, research conducted in Mekelle-Ethiopia [30], Dila-Ethiopia [32], Addis Ababa-Ethiopia [40], Sudan [44], Serbia [48], Sweden [47], Italy [50], Iraq [51], and Germany [52] has been limited to patients with T1DM, excluding those with T2DM. In contrast, our study includes both T1DM and T2DM patients, which could explain the lower

**Table 5. Laboratory findings of diabetic mellitus patients at the UoGCSH, Northwest Ethiopia, from March 1 to September 30, 2021.**

| Characteristics of study participants | Category | Frequency N (%) | Diabetic ketoacidosis | | Fisher's Exact Test |
|---|---|---|---|---|---|
| | | | Positive N (%) | Negative N (%) | P-value |
| Fasting blood glucose levels | ≤250 | 359 (88.6) | 0 (0) | 359 (100.0) | <0.001* |
| | >250 | 46 (11.4) | 35 (76.1) | 11 (23.9) | |
| Urine pH | <7 | 384 (94.8) | 35 (9.1) | 349 (90.9) | 0.201 |
| | ≥7 | 21 (5.2) | 0 (0) | 21 (100.0) | |
| Serum bicarbonate (HCO₃⁻) | <15 mEq/L | 62 (15.3) | 35 (56.5) | 27 (43.5) | 0.423 |
| | ≥15 mEq/L | 343 (84.7) | 0 (0) | 343 (100.0) | |
| Serum anion gap | ≤12 mEq/L | 355 (87.7) | 0 (0) | 355 (100.0) | 0.138 |
| | >12 mEq/L | 50 (12.3) | 35 (70.0) | 15 (30.0) | |
| Urine ketones | Negative | 338 (83.5) | 0 (0) | 338 (100.0) | 0.015* |
| | +1 | 10 (2.5) | 0 (0) | 10 (100.0) | |
| | +2 | 30 (7.4) | 18 (60.0) | 12 (40.0) | |
| | +3 | 15 (3.7) | 10 (66.7) | 5 (33.3) | |
| | +4 | 12 (3.0) | 7 (58.3) | 5 (41.7) | |
| Urine glucose | Negative | 285 (70.4) | 0 (25.0) | 285 (100.0) | 0.062 |
| | +1 | 24 (5.9) | 2 (8.3) | 22 (91.7) | |
| | +2 | 27 (6.7) | 3 (11.1) | 24 (88.9) | |
| | +3 | 28 (6.9) | 10 (35.7) | 18 (64.3) | |
| | +4 | 41 (10.1) | 20 (48.8) | 21 (51.2) | |
| Urine protein | Negative | 280 (70.6) | 10 (3.6) | 270 (96.4) | 0.148 |
| | +1 | 54 (13.3) | 14 (25.9) | 40 (74.1) | |
| | +2 | 37 (9.1) | 5 (13.5) | 32 (86.5) | |
| | +3 | 25 (6.2) | 4 (16.0) | 21 (84.0) | |
| | +4 | 9 (2.2) | 2 (22.2) | 7 (77.8) | |

* The observed difference is statistically significant at P-value < 0.05

prevalence of DKA in our study compared to the higher rates observed in these regions. Moreover, discrepancies in DKA prevalence can result from factors such as variations in study populations, the local prevalence of diabetes, family history of diabetes type, socioeconomic conditions, and delays in diagnosis and treatment [32].

In the present study, DKA was more prevalent in patients with T1DM (70.6%) than in T2DM (29.4%). The same trend is documented in studies conducted in Debre Tabor-Ethiopia (78.8% vs 21.2%) [13], Woldiya-Ethiopia (60% vs 40%) [28], Shashemene-Ethiopia (62.7%, 12.4%) [53], Adama-Ethiopia (68% vs 61%) [31], Jimma-Ethiopia (68.9% vs 31.1%) [36], Hawassa-Ethiopia (28.2% vs 11.8%) [27], Addis Ababa-Ethiopia (87.1% vs 12.9%) [54], Rwanda (74% vs 28.5%) [55], South Africa (61.0% vs 39.0%) [56], and the United Kingdom (75.9% vs 24.1) [57], where a higher prevalence of DKA was observed in cases of T1DM than in T2DM, respectively. A complete insulin deficiency can explain this in patients with T1DM, which triggers lipolysis and results in DKA. In contrast, individuals with T2DM usually retain some residual endogenous insulin, which helps to prevent the formation of lipolysis and DKA. However, they may still develop DKA during periods of stress when insulin deficiency becomes more significant [58, 59]. On the other hand, our findings differ from studies in Burkina Faso (73.3% vs 26.7%) [26], Ghana (51.4% vs 11.4%) [59], Gambia (60.9% vs 8.7%) [41], Thailand (54.8% vs 9.7%) [60], Malaysia (51.5% vs 36.4%) [22], India (80% vs 20%) [61], and China (67.7% vs 15.6%) [62], where DKA was more frequently observed in individuals with T2DM than in T1DM, respectively. The very high number of T2DM and limited number of

**Table 6. Distribution of laboratory findings by diabetes type among patients at the UoGCSH, Northwest Ethiopia, from March 1 to September 30, 2021.**

| Characteristics of study participants | Category | Frequency N (%) | DM type | | Fisher's Exact Test |
|---|---|---|---|---|---|
| | | | T1DM N (%) | T2DM N (%) | |
| Fasting blood glucose levels | ≤250 | 359 (88.6) | 119 (33.1) | 240 (66.9) | **<0.001**\* |
| | >250 | 46 (11.4) | 33 (71.7) | 13 (28.3) | |
| Urine pH | <7 | 384 (94.8) | 146 (38.0) | 238 (62.0) | 0.490 |
| | ≥7 | 21 (5.2) | 6 (28.6) | 15 (71.4) | |
| Serum bicarbonate (HCO₃⁻) | <15 mEq/L | 35 (15.3) | 25 (71.4) | 10 (28.6) | **<0.001**\* |
| | ≥15 mEq/L | 370 (84.7) | 127 (34.3) | 243 (65.7) | |
| Serum anion gap | ≤12 mEq/L | 369 (87.7) | 127 (34.4) | 242 (65.6) | **<0.001**\* |
| | >12 mEq/L | 36 (12.3) | 25 (69.4) | 11 (30.6) | |
| Urine ketones | Negative | 366 (83.5) | 124 (33.9) | 242 (66.1) | **<0.001**\* |
| | +1 | 2 (2.5) | 2 (100.0) | 0 (0) | |
| | +2 | 20 (7.4) | 12 (60.0) | 8 (40.0) | |
| | +3 | 15 (3.7) | 12 (80.0) | 3 (20.0) | |
| | +4 | 2 (3.0) | 2 (100.0) | 0 (0) | |
| Urine glucose | Negative | 285 (70.4) | 91 (31.9) | 194 (68.1) | **0.002**\* |
| | +1 | 24 (5.9) | 6 (25.0) | 18 (75.0) | |
| | +2 | 27 (6.7) | 11 (40.7) | 16 (59.3) | |
| | +3 | 28 (6.9) | 18 (64.3) | 10 (35.7) | |
| | +4 | 41 (10.1) | 23 (56.1) | 18 (43.9) | |
| Urine protein | Negative | 286 (70.6) | 104 (36.4) | 182 (63.6) | 0.930 |
| | +1 | 54 (13.3) | 21 (38.9) | 33 (61.1) | |
| | +2 | 37 (9.1) | 16 (43.2) | 21 (56.8) | |
| | +3 | 25 (6.2) | 10 (40.0) | 15 (60.0) | |
| | +4 | 3 (2.2) | 1 (33.3) | 2 (66.7) | |

\*The observed difference is statistically significant at *P*-value < 0.05

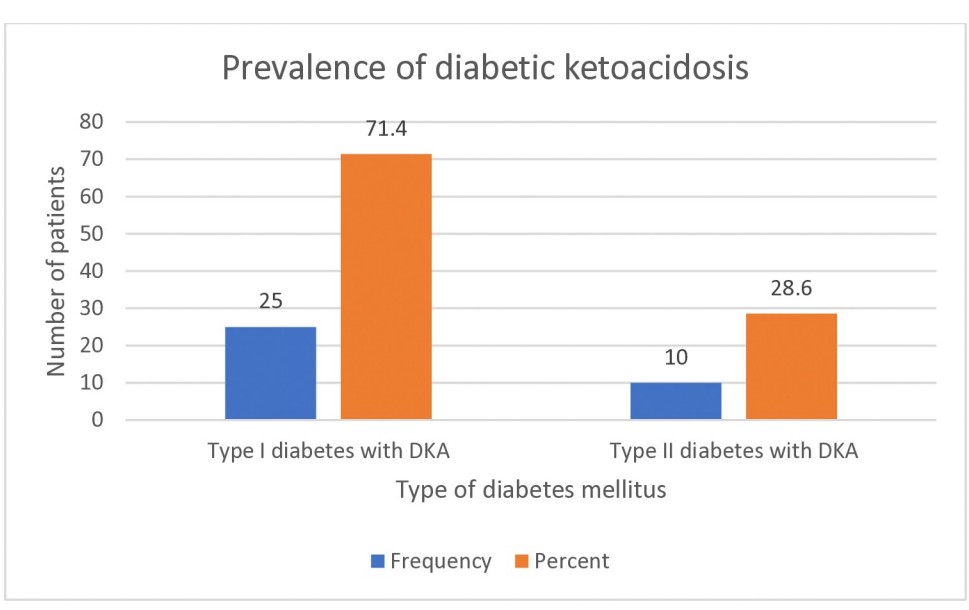

**Fig 2. Prevalence of diabetic ketoacidosis in diabetes mellitus patients at UoGCSH, Northwest Ethiopia, from March 1 to September 30, 2021.**

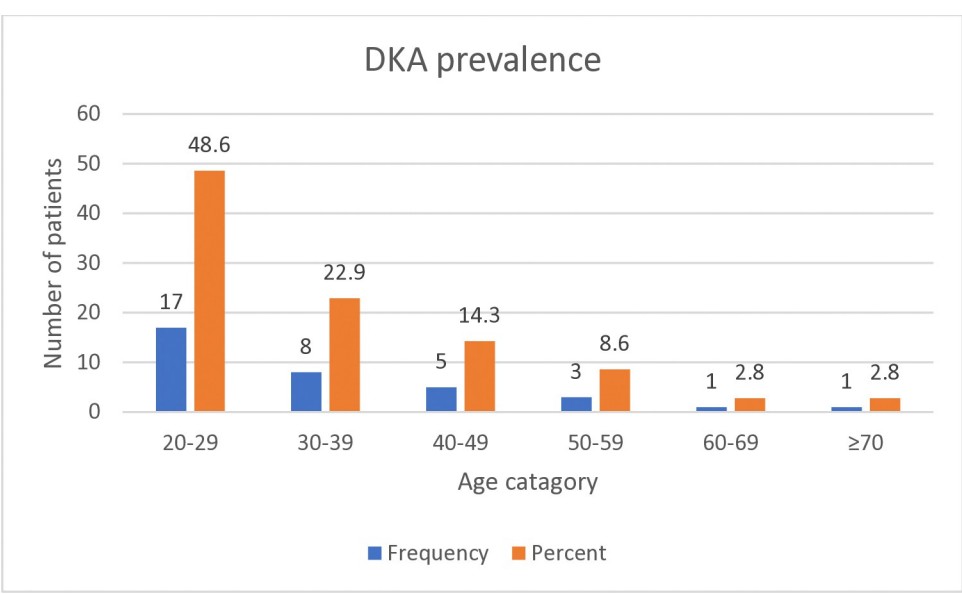

**Fig 3. The prevalence of diabetic ketoacidosis by age group in diabetes mellitus patients at UoGCSH, Northwest Ethiopia, from March 1 to September 30, 2021.**

T1DM in these regions may result in a high proportion of DKA in patients with T2DM. In addition, ethnic or genetic factors in patients from these areas with T2DM might increase their vulnerability to DKA, which cannot be excluded [58].

The present study revealed that the prevalence of DKA is significantly more likely in cases, where the study participants belong to the age group of 20–29 as compared to those whose age is seventy and above (AOR = 2.262; 95%CI = 1.090–4.758; $P$ = 0.013). Comparatively, age is significantly associated with DM complications from a study done in Dessie-Ethiopia ($P$ = 0.048) [29], where study participants in the age group of 31–45 were more likely to develop DM complications than those aged above 45. Similarly, a study in Dila-Ethiopia ($P$ = 0.005) [32] has also reported a high magnitude of DKA among DM patients who were found in the age group of 18–25 as compared to those who were older. Young adults are more prone to developing DKA than older adults due to behavioural, physiological, and clinical factors. Moreover, younger adults with T1DM may have a higher incidence of eating disorders, which can interfere with insulin management and lead to increased risk of DKA [18]. Additionally, the body's counter-regulatory hormone response to insulin deficiency tends to be more pronounced in younger adults, leading to more severe ketosis. Younger adults may also have less experience managing diabetes and recognizing early signs of DKA [63]. Furthermore, beta cell destruction may be more rapid in individuals found in the lower age group than those who are older [30].

The current study presented that DM patients with infections were more likely to develop DKA than those without infections (AOR = 2.819; 95%CI = 1.138–8.428; $P$ = 0.024). This is consistent with the studies conducted in Woldiya-Ethiopia ($P$ = 0.042) [28], Shashemene-Ethiopia ($P$ = 0.00) [53], and Dila-Ethiopia ($P$ = 0.02) [32]. This can be supported by the fact that infection triggers inflammation, the release of pro-inflammatory cytokines, and increases stress hormones, which are counter-regulatory and contribute to insulin resistance, exacerbating DKA. This imbalance between insulin and counter-regulatory hormones drives lipolysis, breaking down triglycerides as an alternative energy source, leading to the production of acidic ketone bodies and resulting in ketonaemia and acidaemia. Once ketone levels surpass the

**Table 7. Bivariable and multivariable logistic regression analysis of factors associated with diabetic ketoacidosis among diabetic mellitus patients at the UoGCSH, Northwest Ethiopia, from March 1 to September 30, 2021.**

| Characteristics of study participants | Category | Diabetic ketoacidosis | | COR (95% CI) | P-value | AOR (95% CI) | P-value |
|---|---|---|---|---|---|---|---|
| | | Yes N (%) | No N (%) | | | | |
| Sex | Male | 25 (12.1) | 182 (87.9) | 1.387 (0.765–3.201) | 0.122 | 1.122 (0.610–2.064) | 0.071 |
| | Female | 10 (5.1) | 188 (94.9) | 1 | 1 | 1 | 1 |
| Age (years) | 20–29 | 17 (32.7) | 35 (67.3) | 1.125 (0.915–3.042) | **0.005**[*] | 2.262 (1.090–4.758) | **0.013**[*] |
| | 30–39 | 8 (13.6) | 51 (86.4) | 1.056 (1.007–5.441) | **0.026**[*] | 1.231 (0.852–6.693) | 0.125 |
| | 40–49 | 5 (7.4) | 63 (92.6) | 1.286 (0.732–4.537) | 0.261 | 1.133 (0.492–2.608) | 0.370 |
| | 50–59 | 3 (2.9) | 99 (97.1) | 1.075 (0.706–7.413) | 0.806 | 0.941 (0.636–7.454) | 0.528 |
| | 60–69 | 1 (1.3) | 78 (98.7) | 1.037 (0.801–5.045) | 0.688 | 1.060 (0.863–7.291) | 0.736 |
| | ≥70 | 1 (2.2) | 44 (97.8) | 1 | 1 | 1 | 1 |
| Residence | Urban | 20 (6.3) | 300 (93.7) | 1 | 1 | 1 | 1 |
| | Rural | 15 (17.6) | 70 (82.4) | 2.311 (0.852–6.638) | **0.001**[*] | 0.551 (0.140–5.710) | 0.441 |
| Marital status | Married | 15 (7.1) | 196 (92.9) | 1 | 1 | 1 | 1 |
| | Single | 7 (8.5) | 75 (91.5) | 1.272 (0.904–5.712) | 0.281 | 0.831 (0.537–3.686) | 0.261 |
| | Divorced | 7 (14.3) | 42 (85.7) | 1.326 (0.626–3.847) | 0.221 | 1.002 (0.606–4.656) | 0.526 |
| | Windowed | 3 (17.6) | 14 (82.4) | 1.254 (1.066–4.979) | **0.047**[*] | 0.768 (0.405–6.694) | 0.407 |
| | Separated | 3 (6.5) | 43 (93.5) | 1.054 (1.010–3.292) | **0.012**[*] | 1.121 (0.841–7.151) | 0.312 |
| Occupation status | Governmental | 2 (3.8) | 50 (96.2) | 1 | 1 | 1 | 1 |
| | Merchant | 8 (11.3) | 63 (88.7) | 1.315 (0.648–6.550) | **0.155**[*] | 1.120 (0.986–5.112) | 0.316 |
| | Housewife | 8 (7.9) | 93 (92.1) | 1.465 (0.870–4.274) | 0.344 | 1.311 (0.641–6.674) | 0.085 |
| | Farmer | 7 (7.1) | 91 (92.9) | 1.520 (0.940–3.599) | 0.426 | 1.434 (0.957–4.103) | 0.624 |
| | Unemployed | 10 (12.0) | 73 (88.0) | 2.292 (1.061–5.390) | **0.022**[*] | 2.578 (1.457–6.113) | **0.017**[*] |
| Income (ETB) | ≤500 | 22 (15.5) | 120 (84.5) | 0.635 (0.550–4.486) | **0.010**[*] | 1.352 (0.951–5.322) | 0.088 |
| | 501–1000 | 7 (8.1) | 79 (91.9) | 0.960 (0.933–6.187) | **0.090**[*] | 0.892 (0.573–3.826) | 0.115 |
| | 1001–1500 | 2 (5.1) | 37 (94.9) | 0.790 (0.552–3.134) | 0.503 | 0.649 (0.383–6.120) | 0.231 |
| | >1500 | 4 (2.9) | 134 (97.1) | 1 | 1 | 1 | 1 |
| Hypertension | Yes | 9 (5.8) | 145 (94.2) | 1.862 (0.848–4.087) | **0.121**[*] | 1.352 (0.901–5.756) | 0.527 |
| | No | 26 (10.4) | 225 (89.6) | 1 | 1 | 1 | 1 |
| Smoking | Yes | 2 (33.3) | 4 (66.7) | 0.810 (0.732–5.022) | **0.053**[*] | 0.995 (0.697–4.519) | 0.418 |
| | No | 33 (8.3) | 366 (91.7) | 1 | 1 | 1 | 1 |
| Alcohol use | Yes | 12 (14.6) | 70 (85.4) | 1.440 (0.812–7.942) | **0.034**[*] | 1.007 (0.816–6.246) | 0.196 |
| | No | 23 (7.1) | 300 (92.9) | 1 | 1 | 1 | 1 |
| Diabetes type | Type I | 25 (16.4) | 127 (83.6) | 1.209 (0.976–3.449) | **0.025**[*] | 3.106 (1.150–7.273) | **0.003**[*] |
| | Type II | 10 (4.0) | 243 (96.0) | 1 | 1 | 1 | 1 |
| Diabetes treatment type | NPH insulin | 25 (16.4) | 127 (83.6) | 1.523 (0.504–4.624) | **0.124**[*] | 1.970 (0.868–9.240) | 0.363 |
| | OHA | 8 (3.5) | 222 (96.5) | 0.854 (0.401–6.074) | **0.136**[*] | 0.492 (0.274–6.355) | 0.601 |
| | NPH and OHA | 2 (8.7) | 21 (91.3) | 1 | 1 | 1 | 1 |
| Presence of infection | Yes | 6 (3.9) | 149 (96.1) | 3.259 (1.321–5.041) | **0.010**[*] | 2.819 (1.138–8.428) | **0.024**[*] |
| | No | 29 (11.6) | 221 (88.4) | 1 | 1 | 1 | 1 |
| Fasting blood glucose | ≤250 | 0 (0) | 359 (100) | 1 | 1 | 1 | 1 |
| | >250 | 35 (76.1) | 11 (23.9) | 2.561 (0.876–7.648) | **0.142**[*] | 1.025 (0.735–9.174) | 0.194 |
| Urine ketones | Negative | 0 (0) | 338 (100) | 1 | 1 | 1 | 1 |
| | +1 | 0 (0) | 10 (100) | 0.964 (0.685–5.763) | 0.441 | 0.653 (0.379–6.541) | 0.624 |
| | +2 | 18 (60.0) | 12 (40.0) | 1.396 (0.982–4.638) | 0.235 | 0.979 (0.708–8.396) | 0.471 |
| | +3 | 10 (66.7) | 5 (33.3) | 1.731 (0.869–6.371) | **0.157**[*] | 1.035 (0.873–5.489) | 0.182 |
| | +4 | 7 (58.3) | 5 (41.7) | 2.475 (1.158–7.081) | **0.130**[*] | 1.217 (0.923–7.590) | 0.093 |

[*] = Significant at P-value ≤ 0.2 for bivariate and P-value < 0.05 for multivariate in regression model adjusted for age, occupation, diabetes type, and presence of infection, AOR = adjusted odds ratio, CI = confidence interval, COR = crude odds ratio, OHA = oral hypoglycemic agents, NPH = neutral protamine Hagedorn

capacity of intracellular buffers, they are excreted in the urine, causing ketonuria [40, 59]. Furthermore, infections may obscure early diabetes symptoms, potentially accelerating the onset of complications [43].

This study found that being unemployed in occupation was identified as a significant risk factor for developing DKA (AOR = 2.578; 95%CI = 1.457–6.113; $P$ = 0.017). This finding is comparable with the finding of Saudi Arabia (P = 0.004) [64]. The association may stem from the fact that unemployment can lead to unhealthy lifestyle choices such as poor diet, lack of physical activity, increased smoking and alcohol consumption, financial difficulties, and inadequate nutrition, all of which can contribute to poor glucose control and deteriorating health, leading to complications [24, 65]. Moreover, individuals with lower incomes may experience delays in accessing healthcare, which can worsen conditions like DKA. Psychological stress associated with financial hardship could further impair glycemic control, increasing the risk of complications [66].

The findings of this study indicate that patients with T1DM are at significantly greater risk of developing DKA compared to those with T2DM (AOR = 3.106; 95% CI = 1.150–7.273; $P$ = 0.003). These results are consistent with studies from Ethiopia, including those conducted in Woldiya and Waghimra (AOR = 2.01) [67], Dessie ($P$ = 0.000) [29], Debre Markos ($P$ = 0.02) [66], and Jimma ($P$<0.001) [38]. The higher susceptibility of T1DM patients to DKA is primarily due to their lack of insulin, which leads to lipid breakdown and ketone production. In contrast, T2DM patients typically retain some endogenous insulin, which inhibits lipolysis and reduces the likelihood of DKA [66, 67].

## Limitations of the study

While the study provides important insights into the prevalence and associated factors of DKA among diabetic patients, it did not include blood gas analysis due to feasibility constraints. Furthermore, the study's cross-sectional design limits the ability to establish causal relationships between the identified factors and the development of DKA.

## Conclusions and recommendations

The prevalence of DKA among diabetic patients receiving follow-up care at UoGCSH was notably high, with a higher occurrence in those with T1DM. Being young adults in age, unemployment, and presence of infection were risk factors to develop DKA in DM patients. Therefore, increased vigilance and regular monitoring of diabetic patients, especially those aged 20–29 years, are crucial for early detection and prevention of DKA. Moreover, prompt identification and treatment of infections in diabetic patients can reduce the risk of DKA. Educating patients on the importance of managing infections early is essential. Furthermore, social and financial support programs for unemployed diabetic patients should be implemented, as unemployment was found to be significantly associated with DKA. This can help improve access to proper diabetes care and reduce stressors contributing to poor glycemic control. In addition, comprehensive diabetes education programs that emphasize the recognition of DKA symptoms and its risk factors should be provided, particularly for those with T1DM and younger patients. Furthermore, future research should include blood gas analysis to improve diagnostic accuracy and provide detailed insights into the biochemical changes associated with DKA. Studies involving multiple centers should also be carried out to ensure diverse population representation, enabling more generalizable findings and addressing regional variations in DKA prevalence and risk factors.

## Acknowledgments

The authors would like to express their deep gratitude to the Department of Clinical Chemistry at the School of Biomedical and Laboratory Sciences, College of Medicine and Health Sciences, University of Gondar, for approving the ethical clearance necessary for this study. Our sincere thanks also go to the Clinical Chemistry staff at the UoGCSH for their support. Special recognition is given to Abel Endesew, Eden Ketema, Mesafint Zeru, and Siyoum Ayele for their contributions during the data collection phase. Lastly, we are thankful to all the study participants.

## Author Contributions

**Conceptualization:** Abebe Birhanu, Sintayehu Ambachew, Netsanet Baye, Emiyamrew Getnet, Eshet Gebrie, Abebaw Worede.

**Data curation:** Abebe Birhanu, Sintayehu Ambachew, Netsanet Baye, Emiyamrew Getnet, Sintayehu Admas, Eshet Gebrie, Abebaw Worede.

**Formal analysis:** Abebe Birhanu, Sintayehu Ambachew, Netsanet Baye, Emiyamrew Getnet, Sintayehu Admas, Abebaw Worede.

**Funding acquisition:** Abebe Birhanu.

**Investigation:** Abebe Birhanu, Sintayehu Ambachew, Netsanet Baye, Abebaw Worede.

**Methodology:** Abebe Birhanu, Sintayehu Admas, Abebaw Worede.

**Project administration:** Abebe Birhanu.

**Resources:** Abebe Birhanu.

**Software:** Abebe Birhanu, Sintayehu Ambachew, Netsanet Baye, Emiyamrew Getnet, Sintayehu Admas, Eshet Gebrie.

**Supervision:** Sintayehu Ambachew, Abebaw Worede.

**Validation:** Abebe Birhanu, Sintayehu Ambachew, Abebaw Worede.

**Visualization:** Abebe Birhanu, Sintayehu Ambachew, Abebaw Worede.

**Writing – original draft:** Abebe Birhanu, Sintayehu Ambachew, Abebaw Worede.

**Writing – review & editing:** Abebe Birhanu, Sintayehu Ambachew, Netsanet Baye, Emiyamrew Getnet, Sintayehu Admas, Eshet Gebrie, Abebaw Worede.

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
