## [Decision Letter · Decision Letter 0]

25 Nov 2024

PONE-D-24-47589Prevalence and associated factors of diabetic ketoacidosis among patients with diabetes mellitus at the University of Gondar Comprehensive and Specialized Referral Hospital Northwest , EthiopiaPLOS ONE

Dear Dr. Birhanu,

Thank you for submitting your manuscript to PLOS ONE. After careful consideration, we feel that it has merit but does not fully meet PLOS ONE’s publication criteria as it currently stands. Therefore, we invite you to submit a revised version of the manuscript that addresses the points raised during the review process.

We look forward to receiving your revised manuscript.

Kind regards,

Timotius Ivan Hariyanto, M.D.

Academic Editor

PLOS ONE

**Journal Requirements:**

Reviewers' comments:

Reviewer's Responses to Questions

**Comments to the Author**

1. Is the manuscript technically sound, and do the data support the conclusions?

Reviewer #1: Partly

Reviewer #2: Yes

2. Has the statistical analysis been performed appropriately and rigorously? 

Reviewer #1: No

Reviewer #2: I Don't Know

3. Have the authors made all data underlying the findings in their manuscript fully available?

Reviewer #1: Yes

Reviewer #2: No

4. Is the manuscript presented in an intelligible fashion and written in standard English?

Reviewer #1: Yes

Reviewer #2: No

5. Review Comments to the Author

**Reviewer #1: **The study has been conducted following the appropriate methodology for epidemiological research and is well designed to meet the predefined objectives. However, the researchers have failed to incorporate into the analysis the interdependency between different patient characteristics, resulting in possibly misleading conclusions. More specifically, the authors highlight in the conclusions the association of DKA with age 30-39, unemployment and presence of infection. For some reason, they omit the conclusion regarding the statistically significant AOR difference between T1D and T2D (Table 4), despite spending a significant portion of discussion on this topic. Knowing that DKA risk is higher in T1D and having that as a significant finding in the study, that should have been the primary observation. Given the difference in pathophysiology and treatment of T1D and T2D and based on the observed significant difference in DKA risk, the rest of the characteristics should have been analyzed separately between these two diagnosis categories. For example, T1D patients are younger than T2D patients. Blending the 2 groups together and showing that younger patients have higher association with DKA warrants a question whether is is a separate clinically relevant characteristic or is it just that they have already observed the correlation with T1D and these patients are younger, hence, younger age has no independent association with DKA. Same is true for the association with infections. It is reasonable to assume that any moderate to severe illness can trigger metabolic instability, possibly resulting in DKA. Such conditions may include decompensation of congestive heart failure, respiratory insufficiency, acute myocardial infraction etc. T1D patients, due to the younger age, do not have high incidence of such comorbidities with infections being the most frequent driver of severe illness. Therefore, it is unclear if infections are an independently associated characteristic or is it T1D with severe illness, which happens to be an infection in this population. The analysis should then look at severe comorbidities in T2D and define if there are other characteristics associated with DKA in T2D patients. All this to say is that while the study provides helpful epidemiological information, further analysis, separating the impact of T1D vs. T2D diagnosis on other associated variables is warranted to allow for clinically relevant conclusions from this study.

**Reviewer #2: **The article assesses the prevalence of DKA in diabetes patients visiting a hospital in Ethiopia over a given time period. The study also determines the significance of several variables as risk-factors for DKA. The article is interesting, but needs further clarification in order to be evaluated for its scientific validity.

Major:

1. The language is not always correct, particularly in the introduction and methods sections. A language editor should be consulted during revision.

2. It is diffucult to understand the study design. Are all patients invited for participation to the study admitted to the hospital for treatment, or are some there for a out-patient checkup? Is the DKA diagnosed upon the visit where the patient is included in the study, or could it occur later within a certain timescope? A more structured step-by-step study procedure should be added to the manuscript. A study flow-chart or timeline should be included in the revised manuscript.

3. The clinical variables (predicting variables in the regression model) should be presented in the methods section.

Minor:

1. The figures are confusing as they mix absolute and relative values. Remove absolute values and only present percent-values. For fig 1, focus on T1D and T2D as they add to 100%.

2. The conclusion in the abstract should be condensed.

3. The conclusion and recommendations should detail what future studies should focus on.

4. The introduction gives a nice introduction to what DKA is, but stops at the production of ketones. How this translates to disease and acidosis should also be added (1 sentence).

5. Is this the prevalence of DKA in the diabetic population, or the prevalence of DKA among patients in need for emergency care at the hospital? Please clarify.

6. The variables used to adjust the regression model in table 4 should be added to the legend of the table.

6. PLOS authors have the option to publish the peer review history of their article (what does this mean?). If published, this will include your full peer review and any attached files.

Reviewer #1: **Yes: **Stanislav Glezer

Reviewer #2: No

---

## [Author Response · Author response to Decision Letter 0]

3 Jan 2025

Rebuttal Letter

Date: January 3, 2025

Original Manuscript Number: PONE-D-24-47589

Original Article Title: Prevalence and associated factors of diabetic ketoacidosis among patients with diabetes mellitus at the University of Gondar Comprehensive and Specialized Referral Hospital Northwest, Ethiopia

To: PLOS ONE

Re: Response to reviewers

Dear Editors,

Thank you for allowing a resubmission of our manuscript, with an opportunity to address the reviewers’ comments.

We are uploading 

(1) Our point-by-point response to the comments below (Response to reviewers)

(2) A marked-up copy with track changes (Revised Manuscript with Track Changes)

 (3) An unmarked version without track changes (Manuscript)

Kind regards,

<Abebe Birhanu> et al

NOTE: We have further addressed the Journal Requirements as follows:

Comment 1. When submitting your revision, we need you to address these additional requirements. Please ensure that your manuscript meets PLOS ONE's style requirements, including those for file naming. The PLOS ONE style templates can be found at 

Authors’ response: We have adhered to PLOS ONE's style and format requirements in the revised manuscript.

Authors Response to Reviewers Comments

Dear reviewers, we sincerely appreciate your valuable, constructive, and prompt feedback. To streamline the review process, we have included your comments below, accompanied by our detailed responses. Each question and concern have been addressed thoroughly, as outlined below: 

Reviewer #1 comment: The study has been conducted following the appropriate methodology for epidemiological research and is well-designed to meet the predefined objectives. However, the researchers have failed to incorporate into the analysis the interdependency between different patient characteristics, resulting in possibly misleading conclusions. More specifically, the authors highlight in the conclusions the association of DKA with age 30-39, unemployment and presence of infection. For some reason, they omit the conclusion regarding the statistically significant AOR difference between T1D and T2D (Table 4), despite spending a significant portion of discussion on this topic. Knowing that DKA risk is higher in T1D and having that as a significant finding in the study, that should have been the primary observation. Given the difference in pathophysiology and treatment of T1D and T2D and based on the observed significant difference in DKA risk, the rest of the characteristics should have been analyzed separately between these two diagnosis categories. For example, T1D patients are younger than T2D patients. Blending the 2 groups together and showing that younger patients have higher association with DKA warrants a question whether is is a separate clinically relevant characteristic or is it just that they have already observed the correlation with T1D and these patients are younger, hence, younger age has no independent association with DKA. Same is true for the association with infections. It is reasonable to assume that any moderate to severe illness can trigger metabolic instability, possibly resulting in DKA. Such conditions may include decompensation of congestive heart failure, respiratory insufficiency, acute myocardial infraction etc. T1D patients, due to the younger age, do not have high incidence of such comorbidities with infections being the most frequent driver of severe illness. Therefore, it is unclear if infections are an independently associated characteristic or is it T1D with severe illness, which happens to be an infection in this population. The analysis should then look at severe comorbidities in T2D and define if there are other characteristics associated with DKA in T2D patients. All this to say is that while the study provides helpful epidemiological information, further analysis, separating the impact of T1D vs. T2D diagnosis on other associated variables is warranted to allow for clinically relevant conclusions from this study.

Authors Response: We truly value your time and effort in reviewing our manuscript. Your feedback has highlighted critical areas that require further attention, and we have taken the necessary steps to address your concerns comprehensively. Upon revisiting our data and analysis, we identified an error in transcribing the number of DKA cases for the first two rows of the age groups in diabetes mellitus patients. This error has been corrected, and the data have been reanalyzed accordingly. Please check lines 43-49, 231-232, 239-240, 247-248, 253-254, 257-258, 262-263, 271-272, 277-289, 292-293, and 376-383.

Reviewer #2 comment: The article assesses the prevalence of DKA in diabetes patients visiting a hospital in Ethiopia over a given time period. The study also determines the significance of several variables as risk-factors for DKA. The article is interesting, but needs further clarification in order to be evaluated for its scientific validity.

Authors Response: We sincerely appreciate the time and effort you have dedicated to reviewing our manuscript and are grateful for your positive feedback.

Reviewer #2 major comment: The language is not always correct, particularly in the introduction and methods sections. A language editor should be consulted during revision.

Authors Response: Thank you for highlighting this issue. We have improved the language usage in the introduction and methods sections of the revised manuscript with the help of local editors. Please check lines 58-223.

Reviewer #2 major comment: It is difficult to understand the study design. Are all patients invited for participation to the study admitted to the hospital for treatment, or are some there for a out-patient checkup? Is the DKA diagnosed upon the visit where the patient is included in the study, or could it occur later within a certain timescope? A more structured step-by-step study procedure should be added to the manuscript. A study flow-chart or timeline should be included in the revised manuscript.

Authors Response: This is really interesting and cornerstone comment. We thank you immensely for this again. We have tried to observe the original manuscript carefully and stated in the revised manuscript clearly like this “The source population for the study were all diabetic patients who had regular follow-up at the diabetes clinic of UoGCSH, while the study population were all diabetic patients who visited the diabetes clinic of UoGCSH during the study period. The study included all adult diabetic patients aged 18 and above who had regular follow-up at the diabetes clinic of UoGCSH. However, pregnant and breastfeeding women were excluded from the study.” Please check lines 119-123. Moreover, our study was a cross-sectional study, which is a snapshot of a population and point prevalence study, is used to analyze data collected at a single point in time. Therefore, in our case, we assessed the prevalence of DKA at the time of diagnosis upon the visit where the patient was included in the study. That means the patient was included only once in this study. Furthermore, we have incorporated the study’s flow-chart or timeline in the revised manuscript and here below based on your valuable suggestion. 

Probability proportional to sample size (proportional allocation technique) was obtained by using the formula: 

Nf = Average no of patients in each Month × total sample size 

 Total follow-up patients in the diabetes clinic 

Number of diabetes patients in March (n = 1400 x 405/ 8400 = 67)

Number of diabetes patients in April (n = 1200 x 405/ 8400 = 58)

Number of diabetes patients in May (n = 1300 x 405/ 8400 = 63)

Number of diabetes patients in June (n = 1200 x 405/ 8400 = 58)

Number of diabetes patients in July (n = 1100 x 405/ 8400 = 53)

Number of diabetes patients in August (n = 1000 x 405/ 8400 = 48)

Number of diabetes patients in September (n = 1200 x 405/ 8400 = 58)

A systematic random sampling technique was used to include the study participants by calculating the kth value, where N= 8400 (based on the average data obtained from seven months of previous follow-up of the diabetes patients). Then, based on the year 2020 within seven consecutive months (March to September 2021) from the UoGCSH diabetes clinic, the sample size was recruited based on this technique and the study participants were included in every Kth value. 

Kth value in March = 1400 / 67= 20.8≈21

Kth value in April = 1200 / 58= 20.7≈21

Kth value in May = 1300 / 63= 20.6≈21

Kth value in June = 1200 / 58= 20.7≈21

Kth value in July = 1100 / 53= 20.8≈21

Kth value in August = 1000 /48= 20.8≈21

Kth value in September n = 1200 / 58= 20.7≈21

Figure 1: The diagrammatic presentation of the step-by-step sampling procedure and patient flow of diabetes patients in the diabetes clinic of UoGCSH, Northwest Ethiopia

Reviewer #2 major comment: The clinical variables (predicting variables in the regression model) should be presented in the methods section.

Authors Response: Thank you again for the valuable comment. We have added the clinical variables (predicting variables) in the method section of the revised manuscript. Please check lines 170-177. 

Reviewer #2 minor comment: The figures are confusing as they mix absolute and relative values. Remove absolute values and only present percent-values. For fig 1, focus on T1D and T2D as they add to 100%.

Authors Response: Dear reviewer, we appreciate your comment. Regarding Figure 1, we have revised it to focus on T1D and T2D exclusively, as they sum to 100%, thereby aligning with your suggestion for clarity. However, we believe that presenting both absolute and relative values together provides a comprehensive view for readers by offering context (percentages) while preserving the raw data (absolute numbers). Absolute values are critical for illustrating the actual burden of DKA cases, while relative values facilitate comparisons between groups. As both metrics are inherently linked (percentages are derived from absolute numbers), presenting them side by side minimizes potential misinterpretation and enhances transparency. Moreover, for clarity, the percentage values in Figures 2 and 3 were calculated by dividing the observed diabetic ketoacidosis (DKA) cases in each group by the total number of DKA cases, as described in the manuscript. Furthermore, we have put legend at the bottom of figures to avoid confusion that signifies the frequency and percentage of DKA across each category. Please check figures in the submission system.

Reviewer #2 minor comment: The conclusion in the abstract should be condensed.

Authors Response: As you requested, we have tried to condense and make necessary corrections in the abstract part of the revised manuscript. Please refer to lines 51-58.

Reviewer #2 minor comment: The conclusion and recommendations should detail what future studies should focus on.

Authors Response: We have forwarded some recommendations regarding how future studies should focus and be conducted to overcome our limitations. Please check lines 401-405. 

Reviewer #2 minor comment: The introduction gives a nice introduction to what DKA is, but stops at the production of ketones. How this translates to disease and acidosis should also be added (1 sentence).

Authors Response: We appreciate your careful observation and this is a valid comment. We have added this text “The accumulation of these acidic ketones causes ketonemia and metabolic acidosis, hallmark features of DKA. Elevated blood glucose levels also trigger osmotic diuresis, resulting in fluid and electrolyte loss. Without timely rehydration, this dehydration impairs renal perfusion, which can reduce the glomerular filtration rate, further exacerbating the metabolic derangements of DKA” to the revised manuscript. Please refer to lines 84-87.

Reviewer #2 minor comment: Is this the prevalence of DKA in the diabetic population, or the prevalence of DKA among patients in need for emergency care at the hospital? Please clarify.

Authors Response: Thank you for pointing out the issue. Our objective was to assess the prevalence of DKA in the diabetic population who had regular follow-ups at the UoGCSH diabetes clinics.

Reviewer #2 minor comment: The variables used to adjust the regression model in Table 4 should be added to the legend of the table.

Authors Response: We have accepted your comment and made necessary corrections accordingly. Please check lines 293-295.

---

## [Decision Letter · Decision Letter 1]

22 Jan 2025

Prevalence and associated factors of diabetic ketoacidosis among patients with diabetes mellitus at the University of Gondar Comprehensive and Specialized Referral Hospital Northwest , Ethiopia

PONE-D-24-47589R1

Dear Dr. Birhanu,

We’re pleased to inform you that your manuscript has been judged scientifically suitable for publication and will be formally accepted for publication once it meets all outstanding technical requirements.

Kind regards,

Timotius Ivan Hariyanto, M.D.

Academic Editor

PLOS ONE

Additional Editor Comments (optional):

Reviewers' comments:

Reviewer's Responses to Questions

**Comments to the Author**

1. If the authors have adequately addressed your comments raised in a previous round of review and you feel that this manuscript is now acceptable for publication, you may indicate that here to bypass the “Comments to the Author” section, enter your conflict of interest statement in the “Confidential to Editor” section, and submit your "Accept" recommendation.

Reviewer #1: All comments have been addressed

Reviewer #2: All comments have been addressed

2. Is the manuscript technically sound, and do the data support the conclusions?

Reviewer #1: Yes

Reviewer #2: Yes

3. Has the statistical analysis been performed appropriately and rigorously? 

Reviewer #1: Yes

Reviewer #2: Yes

4. Have the authors made all data underlying the findings in their manuscript fully available?

Reviewer #1: Yes

Reviewer #2: No

5. Is the manuscript presented in an intelligible fashion and written in standard English?

Reviewer #1: Yes

Reviewer #2: Yes

6. Review Comments to the Author

Reviewer #1: Thanks for addressing the comments. My only remaining suggestion is to include T1D as a risk factor for DKA in the abstract conclusion to be consistent with the manuscript.

Reviewer #2: (No Response)

7. PLOS authors have the option to publish the peer review history of their article (what does this mean?). If published, this will include your full peer review and any attached files.

Reviewer #1: **Yes: **Stanislav Glezer

Reviewer #2: No

---

## [Editor Report · Acceptance letter]

23 Jan 2025

PONE-D-24-47589R1 

PLOS ONE

Dear Dr. Birhanu, 

I'm pleased to inform you that your manuscript has been deemed suitable for publication in PLOS ONE. Congratulations! Your manuscript is now being handed over to our production team.

Kind regards, 

on behalf of

Dr. Timotius Ivan Hariyanto 

Academic Editor

PLOS ONE